# Unleashing SNNs in Object Detection with Time-Evolving Neuron and Dual-Stream Spiking Attention

## Abstract

Brain-inspired Spiking Neural Networks (SNNs) offer remarkable energy efficiency but still lag behind Artificial Neural Networks (ANNs) in fundamental tasks like object detection, primarily due to the precision bottleneck and limited spatial modeling. To narrow this gap, we propose *SpikeDet*, a fully spiking object detector that redefines both the microscopic neuron model and macroscopic attention mechanism. At its core, the bio-inspired *TE-LIF* neuron, with time-evolving membrane dynamics, enhances representational precision and achieves finer input pattern recognition, while maintaining computational efficiency. Building upon this, the proposed *Dual-Stream Spiking Attention* employs a QV-only design that integrates GlobalMixer and LocalAmplifier modules, facilitating effective spatial semantic modeling with linear complexity. Together, these innovations empower SpikeDet to achieve the state-of-the-art performance across multiple object detection benchmarks with minimal energy consumption. On the widely used COCO dataset, SpikeDet achieves **68.3% mAP@50** and **51.9% mAP@50:95**, setting a new milestone in SNN-based detection and even surpassing several popular ANN models. Extensive ablation studies and evaluations across additional vision tasks further validate the effectiveness and generality of our approach.

## 1 Introduction

Spiking Neural Networks (SNNs), regarded as the third generation of neural networks (Maass, 1997), utilize biologically plausible spiking neurons to process information encoded in spatially and temporally distributed spikes. In contrast to artificial neurons, spiking neurons remain mostly inactive and perform computations only when triggered by sparse spikes. This event-driven paradigm significantly enhances energy efficiency (Caviglia et al., 2014; Zhang et al., 2023b), rendering SNNs particularly promising for deployment in real-world applications.

Object detection is a central task in computer vision with applications in autonomous driving, robotics, surveillance, and medical imaging. Unlike image classification, object detection must identify multiple object instances within an image and accurately localize them through bounding boxes. This dual requirement imposes greater demands on representational capacity, numerical precision, and global spatial reasoning.

Despite their efficiency, SNNs still underperform Artificial Neural Networks (ANNs) in object detection, limiting their practical utility. The gap stems from two key factors: (1) *Precision bottleneck.* Unlike ANNs that exploit continuous activations, SNNs convey information via discrete spike sequences. Most existing SNN models operate on Leaky Integrate-and-Fire (LIF) neurons (Maass, 1997), repeating the identical behavior across timesteps and relying on spike counts to approximate continuous activations (Kim et al., 2020a; Yao et al., 2025a). Such a coarse coding scheme is especially detrimental in object detection which demands fine-grained regression. (2) *Limited spatial modeling.* Current SNN detectors, such as SpikeYOLO (Luo et al., 2024) and MSD (Li et al., 2025), rely on convolutional backbones for efficiency, which limits the receptive fields and hinders their ability to understand complex scenes. While the widely adopted self-attention (Vaswani et al., 2017) could help, existing spiking attention modules compatible with SNNs either incur prohibitive overhead or offer insufficient semantic reasoning capabilities (Yao et al., 2025a; Zhou et al., 2024a).

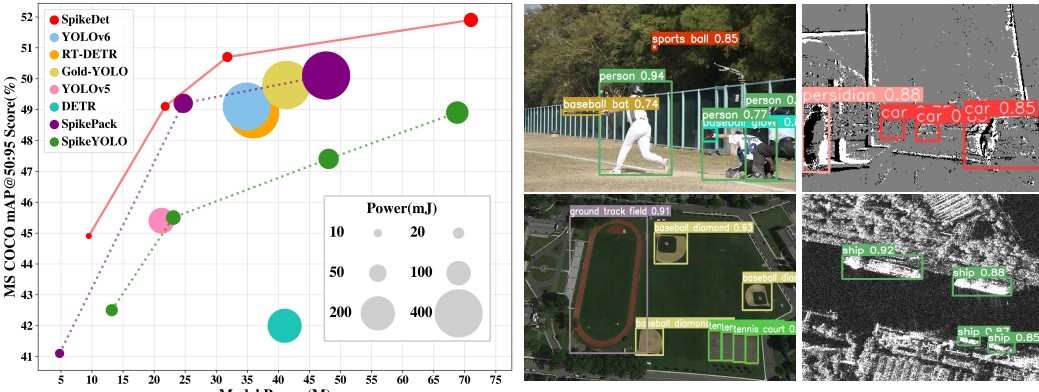

Figure 1: SpikeDet vs. typical object detection models on COCO (left) and its prediction results on COCO, Gen1, NWPU and SSDD (right). SpikeDet achieves SOTA performance among SNNs, surpasses ANNs with superior energy efficiency, and enables strong cross-domain generalization.

To overcome the aforementioned challenges and promote the practical application of SNNs in object detection, we propose a fully spiking object detector—*SpikeDet*, which introduces innovations at both the microscopic level of spiking neurons and the macroscopic level of attention modules.

At the core of our framework is the *Time-Evolving LIF (TE-LIF)* neuron, which extends standard LIF dynamics with time-dependent membrane behavior, inspired by temporal synaptic integration observed in hippocampal neurons (Harris et al., 2002). Unlike vanilla LIF neurons that treat spikes uniformly, TE-LIF assigns different influence to spikes depending on their timing, yielding a finer and more biologically grounded coding scheme. This expands the representational range of spike sequences and supports the precision needed for regression tasks such as bounding box localization. Moreover, the temporal weights are chosen as powers of two, allowing efficient bit-shift implementations that preserve the low-power nature of SNNs while enhancing their expressivity.

To complement this temporal precision with stronger spatial reasoning, we introduce the *Dual-Stream Spiking Attention (DSSA)*. DSSA removes costly matrix multiplications with a query–value (QV)-only design, and uses two coordinated streams—*GlobalMixer* and *LocalAmplifier*—to capture global structure and enhance local details. This design supports long-range feature fusion with linear complexity and maintains stable optimization when combined with TE-LIF.

The proposed *SpikeDet* model achieves the state-of-the-art performance on multiple object detection benchmarks including COCO, Gen1, NWPU, and SSDD. As illustrated in Figure 1, it yields favorable parameter–accuracy trade-offs over leading SNNs, surpasses representative ANNs, reduces energy consumption, and demonstrates robust generalization. Extensive experiments on classification and segmentation further indicate its broad applicability. Our main contributions are as follows:

1) We propose the *TE-LIF* neuron with biologically inspired time-evolving dynamics and power-of-two temporal weights, which enables better representational capacities justified both by our theoretical analysis and empirical experiments while retaining computational efficiency.

2) We introduce *Dual-Stream Spiking Attention*, a QV-only attention mechanism that replaces costly matrix multiplications and couples *GlobalMixer* with *LocalAmplifier* for efficient global–local spatial modeling with linear complexity.

3) We integrate these designs into *SpikeDet*, a spiking detector that achieves SOTA performance across diverse detection domains. SpikeDet attains **68.3% mAP@50** and **51.9% mAP@50:95** on COCO, outperforming the previous best directly trained SNN by **+2.1%** and **+3.0%**, respectively, while consuming only **32.4 mJ**—less than **9%** of YOLOv6 and RT-DETR's energy usage.

## 2 RELATED WORK

**Spiking Neural Networks** (SNNs) employ biologically plausible spiking neurons, enabling event-driven computation with high energy efficiency (Li et al., 2024). However, representing information

through binary spike sequences rather than continuous activations leads to significant information loss, limiting the representational power of SNNs. Current efforts to build high-performance SNNs follow two main directions: (1) ANN-to-SNN conversion (Deng & Gu, 2021; Hu et al., 2023), which approximates ANN activations by spike rates but suffers from high latency and limited adaptability, and (2) direct training with surrogate gradients and spatio-temporal backpropagation (Wu et al., 2018; Neftci et al., 2019), which supports low-latency, end-to-end optimization. To reduce precision loss from discretizing membrane potentials, recent studies adopt integer-valued activations during training and map them to spike counts at inference (Luo et al., 2024; Qiu et al., 2025; Lei et al., 2025). However, these approaches mostly rely on vanilla LIF neurons, which applies identical dynamics across all timesteps and underutilizes the expressive potential of spike sequences. This raises a natural question: *why not endow neurons with time-evolving dynamics?*

**Object Detection** demands precise spatial reasoning and continuous-valued regression, which remain challenging for SNNs. While artificial neural network (ANN) detectors have evolved from two-stage frameworks (Girshick et al., 2014; Girshick, 2015) to real-time, one-stage models such as YOLO (Redmon et al., 2016; Bochkovskiy et al., 2020) and more recent transformer-based architectures like DETR (Carion et al., 2020; Zhu et al., 2020), SNN detectors have historically lagged. Early SNN detectors (Kim et al., 2020a;b; Su et al., 2023) performed poorly, while recent models such as SpikeYOLO (Luo et al., 2024) improved accuracy via integer-valued training, and SpikePack (Shen et al., 2025) achieved competitive results through ANN-to-SNN conversion but at prohibitive energy cost. However, SNN detectors typically adopt convolutional backbones with limited receptive fields, thus struggling to model long-range semantic dependencies. Meanwhile, their neuronal dynamics remain coarse—representations based on monotonous behavior and simple summation constrain expressiveness, posing challenges for dense regression tasks like object detection.

**Attention mechanisms** have proven effective in ANN-based vision models, facilitating global context modeling (Dosovitskiy et al., 2020; Liu et al., 2021b). However, despite some efforts to incorporate self-attention into object detection (Zhao et al., 2024a; Tian et al., 2025), most architectures still depend on CNNs due to the quadratic computational cost of self-attention (Glenn, 2023; Wang et al., 2023a; Li et al., 2023). Recently, attention has also been explored in SNNs. Methods, like SDSA-3 (Yao et al., 2024), mimic vanilla self-attention by relying on matrix multiplication, while other approaches (e.g., SSA, SDSA) simplify it through element-wise operations or summation (Zhou et al., 2022; Yao et al., 2024; 2023a; Zhou et al., 2024a; Deng et al., 2024). However, these modules suffer from either high computational complexity or limited feature mixing. Thus, designing an effective attention mechanism tailored for SNN detection remains an open challenge.

## 3 PRELIMINARY

**LIF.** SNNs exhibit spatio-temporal dynamic properties via biologically inspired spiking neurons, among which the Leaky Integrate-and-Fire (LIF) neuron (Maass, 1997) is the most frequently adopted. At each timestep $t$, LIF neuron repeats the identical dynamics, formally defined as:

$$V_t = \beta H_{t-1} + I_t \; ; \; S_t = \Theta(V_t - V_{th}) \; ; \; H_t = V_t - V_{th} \cdot S_t \qquad (1)$$

The membrane potential $V_t$ integrates spatial input $I_t$ and temporal input $\beta H_{t-1}$, where $H_{t-1}$ is the previous membrane potential and $\beta$ is the decay factor. The Heaviside step function $\Theta(x)$ outputs 1 when $x > 0$ and 0 otherwise. The neuron emits a spike $S_t = 1$ and the residual membrane potential $H_t$ is reduced by $V_{th}$ if $V_t$ exceeds the firing threshold $V_{th}$, else $H_t$ remains equal to $V_t$.

**Self-Attention Mechanism.** For the input $X \in \mathbb{R}^{N \times D}$, where $N$ is the token count and $D$ is the embedding dimension, vanilla self-attention (VSA) (Vaswani et al., 2017) is formulated as follows:

$$Q = W_Q X \; , \; K = W_K X \; , \; V = W_V X \; ; \; \text{VSA}(Q, K, V) = \text{softmax}(\frac{QK^{\mathrm{T}}}{\sqrt{D}})V \qquad (2)$$

However, the reliance on floating-point matrix multiplications and the exponential operations in the softmax function undermines the spike-driven characteristics of SNNs (Yao et al., 2024; Zhou et al., 2024a). Moreover, the $O(N^2 D)$ quadratic complexity of VSA proves impractical for resource-constrained object detection and contradicts the goal of energy efficiency in SNNs.

# 4 METHOD

Our proposed SpikeDet model addresses the performance degradation of SNNs in object detection through co-designs at both the neuron and network module levels. We first introduce the highly expressive TE-LIF spiking neuron as the core component of our framework. Next, we introduce the Dual-Stream Spiking Attention module, which enhances both global and local semantic modeling while maintaining compatibility with TE-LIF.

## 4.1 TIME-EVOLVING SPIKING NEURON

**Limitations of Vanilla LIF.** Traditional LIF neurons apply identical dynamics across the entire time window and assign equal importance to spikes at each timestep. As a result, synaptic strength is encoded simply by counting spikes over $T$ steps, i.e., $\sum_{t=1}^{T} S_t$, yielding at most $T + 1$ discrete levels. Such a coarse representation introduces a precision bottleneck, rendering it inadequate for dense regression tasks like object detection.

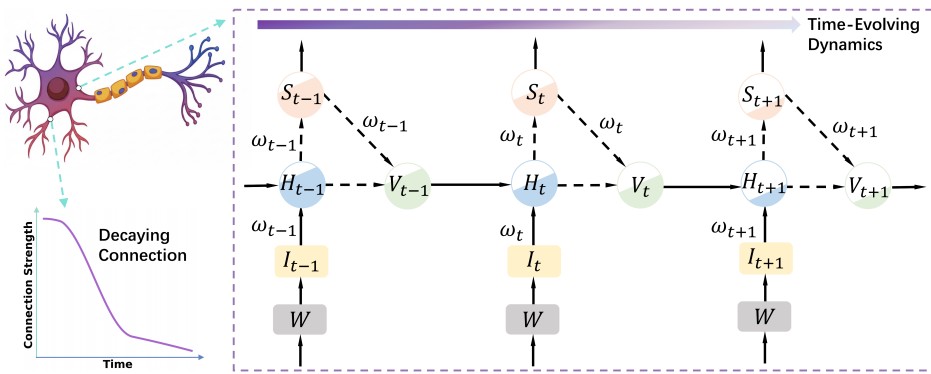

Figure 2: Time-Evolving LIF neuron. To capture the temporal shift of synaptic strength between hippocampal neurons, we extend the traditional LIF neuron model by integrating time-evolving dynamics, modulated by a time-decaying weight $\omega_t$.

**Biological Motivation.** In biological systems, temporal-weighted synaptic integration is common, for example in hippocampal neurons (Harris et al., 2002). The impact of a presynaptic spike varies with timing and context, leading to postsynaptic responses of non-uniform magnitude. Additionally, Harris et al. (2002); Deperrois & Graupner (2020) observe that dendritic connection strength decays with spike timing over the short term. Inspired by these phenomena, we propose the *Time-Evolving LIF (TE-LIF)* neuron. Compared to standard LIF, TE-LIF incorporates dynamic integration magnitude, firing threshold, and decay level, yielding more biologically faithful temporal behavior.

**Mathematical Formulation.** Mathematically, we incorporate a time-decaying weight $\omega_t = 2^{T-t}$ ($t = 1, \ldots, T$) in the TE-LIF neuron during both the charging and discharging phases, as illustrated in Figure 2. This formulation underscores that earlier spikes trigger stronger response and adaptation:

$$V_t = \beta H_{t-1} + I_t \cdot \omega_t \; ; \; S_t = \Theta(V_t - V_{th} \cdot \omega_t) \; ; \; H_t = V_t - V_{th} \cdot S_t \cdot \omega_t \tag{3}$$

This neuronal model enables a finer-grained connection strength $\sum_{t=1}^{T} \omega_t S_t$, which expands the representational range to $2^T$ values and substantially improves precision. Since $\omega_t$ is a power of two, multiplication can be implemented as a bit-shift, providing efficient deployment on digital hardware (Kim et al., 2018; Zhang et al., 2021; 2020). TE-LIF therefore bridges biologically inspired coding with the precision and efficiency required by regression tasks.

**Theoretical Insight.** We analyze the expressivity of TE-LIF using the framework of input space partitioning, which measures how many distinct polyhedral regions a network can form (Pascanu et al., 2013; Montúfar et al., 2014; Nguyen et al., 2025). Each neuron contributes to the partitioning by inducing hyperplanes within the input domain, and a larger number of partitions indicates a stronger capacity for function approximation. We present a theorem showing that TE-LIF achieves a higher complexity of partitions in the input space compared to LIF as the time window $T$ increases.

**Theorem 4.1.** *Consider a shallow discrete-time SNN with $T$ timesteps in the input space $\mathbb{R}^{n_{in}}$. The number of regions partitioned by LIF-SNN and TE-LIF-SNN are respectively bounded by:*

$$N_{LIF}(T) = N_{LIF}(0) + \sum_{t=1}^{T} \Delta N_{LIF}(t) \le 1 + \sum_{t=1}^{T} t = 1 + \frac{T(T+1)}{2} \in O(T^2),$$

$$N_{TE}(T) = N_{TE}(0) + \sum_{t=1}^{T} \Delta N_{TE}(t) \le 1 + \sum_{t=1}^{T} C \cdot t^2 = 1 + C \cdot \frac{T(T+1)(2T+1)}{6} \in O(T^3),$$

*where the constant $C = 1 + \left\lceil \frac{\hat{\beta}-1}{\hat{\beta}+1} + \frac{(\hat{\beta}-1)|H_0|}{(\hat{\beta}+1)V_{th}} \right\rceil$, $H_0$ is the initial membrane potential, $V_{th}$ denotes the firing threshold, and the decay ratio $\hat{\beta} = \frac{\omega_{t-1}}{\omega_t} = 2$ in our setting.*

Remark. Theorem 4.1 (with a detailed proof in Appendix A) indicates that LIF neurons yield $O(T^2)$ partitions, while TE-LIF achieves $O(T^3)$ due to its time-evolving dynamics creating more non-parallel hyperplanes. This cubic growth enables finer input discrimination, which is especially valuable for regression tasks such as bounding box localization.

**Training Stage.** In order to lower temporal redundancy and accelerate the training phase, we utilize the multi-bit training method commonly used in prior SNN studies (Luo et al., 2024; Yao et al., 2025a; Qiu et al., 2025; Lei et al., 2025), which merges several timesteps into one step and allows integer-valued spike while training. To ensure the differentiability of the network, straight-through estimator (Bengio et al., 2013) are appiled for the integer-valued activation function. When inference, the integer values are restored into 0/1 spike sequences via extending virtual timesteps.

**Inference Stage.** During inference, TE-LIF neuron operates as in Equation 3. Each neuron in layer $l-1$ emits a binary spike $S_t^{l-1} \in \{0, 1\}$ at each timestep, and the input to layer $l$ is computed with the synaptic weight matrix $W^l$ between two layers:

$$W^l I_t^l = (W^l S_t^{l-1}) \cdot \omega_t \tag{4}$$

Since $S_t^{l-1}$ is binary, the matrix multiplication $W^l \cdot S_t^{l-1}$ simplifies to sparse masking and accumulation operations, where only a few 1-valued spikes trigger the reading and accumulation of the corresponding weights. Furthermore, since $\omega_t = 2^{T-t}$, the multiplication by $\omega_t$ can be replaced with a lightweight bit-shift, i.e., $x \cdot \omega_t = x \ll (T - t)$. This leads to a highly efficient inference process in which expensive multiply-accumulate (MAC) operations are replaced by few masking and accumulate (AC) operations, preserving the low-power characteristic of SNNs. Moreover, we offer an adder-only implementation of TE-LIF neuron in Verilog hardware description language, as detailed in Appendix B. The use of accumulations, bit-shift operations and sparse events makes our approach compatible with typical neuromorphic chips that support event-based computation (Davies et al., 2018; Furber et al., 2014; Kim et al., 2018; Zhang et al., 2020).

## 4.2 DUAL-STREAM SPIKING ATTENTION

To enhance the spatial modeling capacity of SNN detectors and improves computational efficiency, we design a Dual-Stream Spiking Attention, which enables effective global–local feature fusion and maintains stable optimization when combined with TE-LIF neuron.

**Eliminate Matrix Multiplication.** Vanilla self-attention relies on successive matrix multiplications. However, with TE-LIF neurons, such intensive operations yields values beyond the encoding range, leading to approximation errors and training instability. To address this while boosting efficiency, we replace matrix multiplications with element-wise operations, following Zhai et al. (2021).

**QV-only Design.** We propose a QV-only design that removes the Key branch. This is driven by the fact that QK interactions struggle to measure similarity under sparse, spike-coded activations and tend to amplify noise at low time steps (Wang et al., 2025; Xiao et al., 2025). Instead of computing QK relevance, we directly construct the attention map from Q using a global-local fusion scheme.

**Global-Local Fusion.** Existing spiking attention modules (Zhou et al., 2024a; Yao et al., 2023a; Wang et al., 2025) have primarily relied on element-wise operations that usually fuse features along either the token or the embedding dimension, but rarely both. This limits their ability to capture

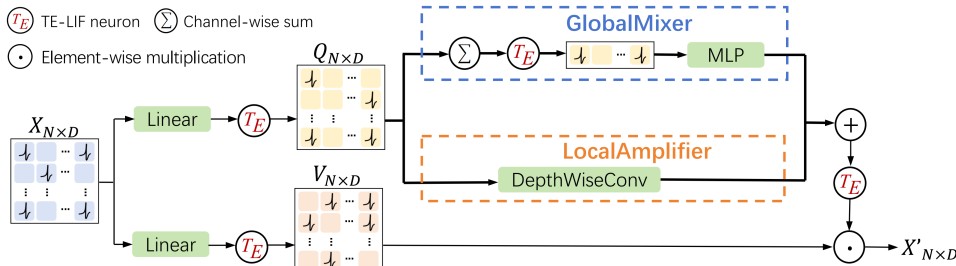

Figure 3: Dual-Stream Spiking Attention. Built on TE-LIF neurons, DSSA removes the Key branch and matrix multiplication, and fuses features through complementary global and local streams.

long-range dependencies under low firing rates or short windows. Inspired by human vision—where saccades provide a global view and fixations allow detailed local analysis (Deubel & Schneider, 1996)—as depicted in Figure 3, we design two complementary components:

$$\text{GlobalMixer}(Q) = \text{MLP}\left(\text{TE}\left(\sum_c Q\right)\right), \quad \text{LocalAmplifier}(Q) = \text{DepthWiseConv}(Q) \quad (5)$$

Here, TE denotes the TE-LIF neuron. GlobalMixer aggregates token-level information through channel summation and then applies a lightweight MLP to capture inter-channel interactions. LocalAmplifier focuses on spatial locality via depthwise convolution without cross-channel mixing. Together, they enable balanced global and local feature modeling with linear complexity.

**Formal Definition of DSSA.** Given the binary input feature $X \in \{0,1\}^{N \times D}$, DSSA operates as follows:

$$Q = \text{TE}(\text{BN}(XW_Q)), \quad V = \text{TE}(\text{BN}(XW_V)) \quad (6)$$

$$Attn = \text{GlobalMixer}(Q) + \text{LocalAmplifier}(Q), \quad X' = \text{TE}(Attn \odot V) \quad (7)$$

**Linear Complexity and Practical Benefits.** DSSA achieves linear complexity $O(ND)$ with respect to both token and channel dimensions and avoids floating-point multiplications. It operates on event-driven accumulations and bit-shift computations, ensuring stable training and leveraging TE-LIF's performance benefits. A more detailed formulation and complexity analysis are provided in Appendix C. In subsection 5.4, we conduct a quantitative and qualitative comparison of DSSA with other SNN attention mechanisms, demonstrating that our design achieves superior performance through efficient global and local fusion.

## 5 EXPERIMENTS

To thoroughly assess our approach, we incorporate the TE-LIF neuron and Dual-Stream Spiking Attention (DSSA) module into the macro architecture of YOLOv12 (Tian et al., 2025), forming the SpikeDet model. We evaluate SpikeDet on various object detection benchmarks, including the frame-based COCO2017 dataset (Lin et al., 2014), the event-based Gen1 dataset (De Tournemire et al., 2020), and two remote-sensing datasets: NWPU (Cheng et al., 2017) and SSDD (Wang et al., 2019). Detailed descriptions of all datasets are provided in Appendix D.

### 5.1 EXPERIMENTAL SETUP

Our models are primarily trained on 8 NVIDIA RTX 4090 GPUs using the SGD optimizer with a learning rate of 0.01 and a timestep setting $T = 8$. Further training details are in Appendix E.

For object detection evaluation, we report the mAP (mean Average Precision) at IoU=0.5 (mAP@50), the average mAP over IoU thresholds from 0.5 to 0.95 (mAP@50:95).

For energy consumption analysis, we adopt the standard protocol widely used in the SNN literature (Panda et al., 2020; Yao et al., 2023b; Yin et al., 2021; Luo et al., 2024). The energy cost of ANNs is calculated as the number of floating-point operations (FLOPs) multiplied by the energy per Multiply-Accumulate operation ($E_{\text{MAC}} = 4.6\text{pJ}$), while that of SNNs is derived by multiplying the FLOPs by the energy per Accumulate operation ($E_{\text{AC}} = 0.9\text{pJ}$) and then scaling the result by the average firing rate (Horowitz, 2014). Additional computational details are provided in Appendix F.

## 5.2 COCO OBJECT DETECTION

We train SpikeDet models of various sizes (S, M, L, X) on the COCO dataset. As detailed in Table 1, SpikeDet achieves the state-of-the-art performance among SNN-based detectors, with an **mAP@50 of 68.3%** and **mAP@50:95 of 51.9%**, surpassing the previous best directly trained SNN by **+2.1%** and **+3.0%** respectively. Compared to the best ANN2SNN detector—SpikePack, SpikeDet attains higher accuracy while consuming only **32.4 mJ**—just **8.1%** of SpikePack's energy usage. Notably, SpikeDet-L outperforms YOLOv6 and RT-DETR, two widely used ANN-based detectors, with a comparable number of parameters and less than **4%** of their power consumption. This significantly narrows the performance gap between SNNs and ANNs in object detection, highlighting the strong potential of SNNs for practical deployment.

Table 1: Performance of object detection on COCO val2017

| Type | Model | Param(M) | Power(mJ) | mAP@50(%) | mAP@50:95(%) |
|---|---|---|---|---|---|
| ANN | ResNet-18 (Yu et al., 2022) | 31.2 | 890.6 | 54.0 | 34.0 |
| | PVT (Wang et al., 2021) | 32.9 | 520.3 | 59.2 | 36.7 |
| | DETR (Carion et al., 2020) | 41.0 | 197.8 | 62.4 | 42.0 |
| | YOLOv5 (Jocher et al., 2020) | 21.2 | 112.5 | 64.1 | 45.4 |
| | RT-DETR (Zhao et al., 2024b) | 36.0 | 460.0 | 66.8 | 48.9 |
| | YOLOv6 (Li et al., 2023) | 34.9 | 394.7 | 66.1 | 49.1 |
| | Gold-YOLO (Wang et al., 2024a) | 41.3 | 402.5 | 67.0 | 49.8 |
| ANN2SNN | Spiking-Yolo (Kim et al., 2020b) | 10.2 | - | - | 25.7 |
| | Bayesian Optim (Kim et al., 2020a) | 10.2 | - | - | 25.9 |
| | Spike Calib (Li et al., 2022) | 17.1 | - | 45.4 | - |
| | SUHD (Qu et al., 2024) | 7.2 | - | 54.6 | - |
| | SpikePack (Shen et al., 2025) | 47.7 | 400.7 | 67.9 | 50.1 |
| Directly Trained SNN | Spiking Retina (Zhang et al., 2023a) | 11.3 | 21.4 | 28.5 | - |
| | EMS-Res-SNN (Su et al., 2023) | 26.9 | 29.0 | 50.1 | 30.1 |
| | Meta-SpikeFormer (Yao et al., 2024) | 75.0 | 140.8 | 51.2 | - |
| | Ensemble SNN (Ding et al., 2025) | 13.2 | - | 54.0 | 38.4 |
| | SpikingYOLOX (Miao et al., 2025) | 7.8 | - | 56.7 | 37.1 |
| | QSD-Transformer (Qiu et al., 2025) | 34.9 | 117.2 | 57.0 | - |
| | E-Spikeformer (Yao et al., 2025a) | 38.7 | 119.5 | 58.8 | - |
| | SpikeYOLO (Luo et al., 2024) | 68.8 | 84.2 | 66.2 | 48.9 |
| | **SpikeDet-S (Ours)** | 9.5 | 4.8 | 61.0 | 44.9 |
| | **SpikeDet-M (Ours)** | 21.8 | 11.8 | 65.5 | 49.1 |
| | **SpikeDet-L (Ours)** | 31.8 | 15.6 | 67.2 | 50.7 |
| | **SpikeDet-X (Ours)** | 71.0 | 32.4 | **68.3 (+2.1)** | **51.9 (+3.0)** |

## 5.3 DVS AND REMOTE SENSING DETECTION

As shown in Table 2, SpikeDet achieves the state-of-the-art performance on the Gen1 dataset in terms of mAP@50:95 among SNN-based models, without any specialized adaptation for DVS (Dynamic Vision Sensor) data, demonstrating strong generalization capability across diverse modalities. SpikeDet also surpasses both ANN and SNN baselines on the NWPU and SSDD remote sensing datasets, confirming its suitability for resource-constrained edge-based aerial imagery applications.

## 5.4 ABLATION STUDY

To assess the effectiveness of our spiking neuron and attention design, we conduct detailed ablation studies on the COCO dataset using our SpikeDet model.

**TE-LIF Analysis.** To validate the representational capability advantage of TE-LIF neurons over standard LIF neurons, we train a SpikeDet-S model by replacing TE-LIF with LIF at $T = 4$, optimized using STBP (Spatial Temporal BackPropagation). As reported in Table 3, under the same time window $T = 4$, the model equipped with TE-LIF delivers significantly higher accuracy.

To assess the effect of time window length $T$ in TE-LIF, we evaluate SpikeDet models under various $T$ settings. As shown in Table 3, increasing $T$ generally improves performance, though gains

Table 2: Performance of object detection on Gen1, NWPU and SSDD

| Dataset | Type | Model | Param(M) | mAP@50(%) | mAP@50:95(%) |
|---|---|---|---|---|---|
| Gen1 | ANN | AEGNN (Schaefer et al., 2022) | 20 | - | 16.3 |
| | | RRC-Events (Chen, 2018) | >100 | - | 30.7 |
| | | RED (Perot et al., 2020) | 24.1 | - | 40.0 |
| | SNN | SpikeFPN (Zhang et al., 2024) | 22 | 47.7 | 22.3 |
| | | Tr-SpikingYolo (Yuan et al., 2024) | 7.9 | 45.3 | - |
| | | SFOD (Fan et al., 2024) | 11.9 | - | 32.1 |
| | | CREST (Mao et al., 2025) | 7.61 | - | 36.0 |
| | | SpikingViT (Yu et al., 2024) | 21.5 | 61.6 | 39.4 |
| | | MSD (Li et al., 2025) | 7.8 | 66.3 | 38.9 |
| | | SpikeYOLO (Luo et al., 2024) | 23.1 | 67.2 | 40.4 |
| | | EAS-SNN (Wang et al., 2024c) | 25.3 | **69.9** | 37.5 |
| | | **SpikeDet (Ours)** | 21.8 | 68.4 | **41.2** |
| NWPU | ANN | ABNet (Liu et al., 2021a) | 27.1 | 87.3 | - |
| | | YOLOv3 (Redmon & Farhadi, 2018) | 58.7 | 82.6 | 52.1 |
| | | YOLOv5-Swin (Liu et al., 2021b) | 13.4 | 89.8 | 53.9 |
| | | $CS^n$Net (Chen et al., 2023) | 12.2 | 90.4 | 55.4 |
| | SNN | EMS-YOLO (Su et al., 2023) | 14.4 | 87.9 | - |
| | | SNN-ViT-YOLO (Wang et al., 2025) | 53.7 | 89.4 | - |
| | | **SpikeDet (Ours)** | 21.8 | **90.5** | **59.3** |
| SSDD | ANN | FasterR-CNN (Fu et al., 2020) | 25.6 | 85.3 | - |
| | | YOLOv3 (Redmon & Farhadi, 2018) | 58.7 | 88.6 | 44.3 |
| | | YOLOv5-Swin (Liu et al., 2021b) | 13.4 | 94 | 57.9 |
| | | Improved PRDet (Yu et al., 2021) | 35.5 | 96.5 | 64.3 |
| | | $CS^n$Net (Chen et al., 2023) | 12.2 | 97.1 | 64.9 |
| | SNN | EMS-YOLO (Su et al., 2023) | 14.4 | 95.1 | - |
| | | SNN-ViT-YOLO (Wang et al., 2025) | 53.7 | 97.0 | - |
| | | **SpikeDet (Ours)** | 21.8 | **98.5** | **75.5** |

saturate beyond $T = 8$. Moreover, SpikeDet achieves strong performance with short time windows ($T = 3$ or $4$). This robustness reflects the high expressivity of TE-LIF neurons and the proposed attention mechanism, underscoring the practicality of our approach for real-world deployment.

Table 3: The blue, red and yellow regions respectively show the impact of neuron types, the effect of time steps, and SpikeDet's performance under short time windows.

Table 4: The blue region demonstrates the impact of architectural modifications, while the red region compares the performance of other typical SNN attention modules.

| Model | $T$ | mAP@50 (%) | mAP@50:95 (%) |
|---|---|---|---|
| SpikeDet-S (LIF) | 4 | 45.7 | 31.0 |
| SpikeDet-S | 4 | 59.9 | 44.2 |
| SpikeDet-S | 8 | 61.0 | 44.9 |
| SpikeDet-S | 10 | 61.0 | 44.9 |
| SpikeDet-M | 4 | 65.4 | 48.9 |
| SpikeDet-L | 3 | 66.1 | 49.5 |
| SpikeDet-L | 4 | 67.2 | 50.5 |
| SpikeDet-X | 4 | 67.6 | 51.0 |

| Attention | mAP@50(%) | mAP@50:95(%) |
|---|---|---|
| **DSSA (Ours)** | **61.0** | **44.9** |
| Remove GM | 60.5 | 44.4 |
| Remove LA | 60.5 | 44.4 |
| Add $K$ Branch | 60.8 | 44.7 |
| SDSA-1 | 57.0 | 41.5 |
| SDSA-3 | 60.2 | 44.2 |
| QKTA | 59.8 | 44.1 |
| QKCA | 60.0 | 43.9 |

**Attention Analysis.** To evaluate the effectiveness of DSSA, we conduct a series of controlled structural modifications based on SpikeDet-S. Specifically, we remove the GlobalMixer (GM) and LocalAmplifier (LA) components individually from DSSA. We also introduce a $K$ branch to restore QK interactions via element-wise multiplication. As reported in Table 4, removing either GM or LA degrades accuracy, suggesting that both local and global cues contribute to performance, though neither dominates alone. Meanwhile, introducing QK interaction provides no meaningful gain, which implies that explicit pairwise token interactions are unnecessary under our spike-driven formulation.

We further replace DSSA with several representative alternatives including SDSA-1 (Yao et al., 2023a), SDSA-3 (Yao et al., 2025b), QKTA, and QKCA (Zhou et al., 2024a). We also test with SSSA(Wang et al., 2025), but its axis-wise summation followed by matrix multiplication produces excessively large values, causing gradient vanishing and training failure. Table 4 shows that our method outperforms all baselines with linear complexity. In terms of practical efficiency, a single inference of DSSA consumes only **36%** of the power required by SDSA-3, a prevalent SNN attention mechanism. Moreover, EigenCAM visualizations in Figure 4 qualitatively demonstrate that DSSA captures more complete and spatially coherent semantic regions.

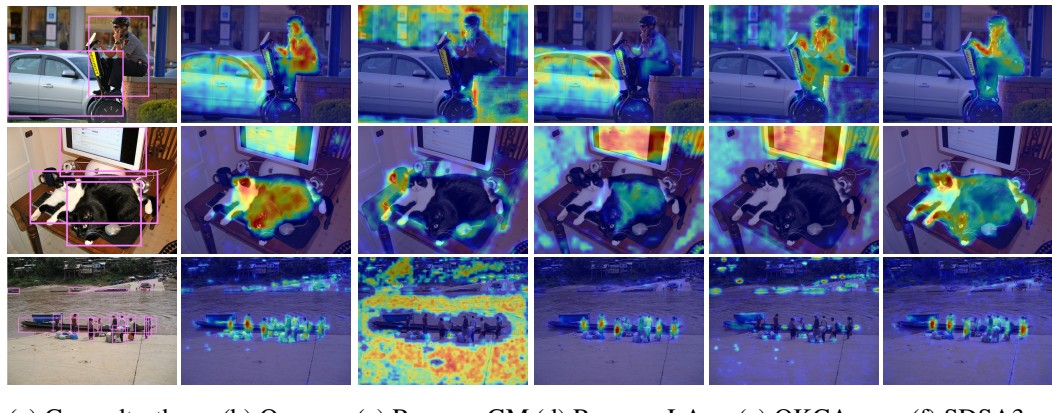

(a) Groundtruth     (b) Ours     (c) Remove GM (d) Remove LA     (e) QKCA     (f) SDSA3

Figure 4: The pink boxes in the input images enclose the objects, while the heatmaps indicate the regions where the attention module focuses, derived from intermediate-layer features.

## 5.5 APPLICATIONS BEYOND OBJECT DETECTION

**ImageNet Classification.** To assess the generalizability of our methods across diverse vision tasks and model architectures, we integrate TE-LIF and DSSA into E-SpikeFormer (Yao et al., 2025a) and evaluate the resulting model on the ImageNet classification benchmark (Deng et al., 2009). As reported in Table 6 of Appendix G, our approach attains a competitive Top-1 accuracy of **79.7%**, surpassing numerous existing models of comparable or even larger scale.

**ADE20K Segmentation.** Semantic segmentation is a challenging computer vision task that requires dense, pixel-wise classification, demanding both fine-grained spatial modeling and comprehensive contextual reasoning across the entire image. We employ our ImageNet-pretrained model as the backbone, add a segmentation head, and fine-tune the network on the ADE20K dataset. As shown in Table 7 of Appendix G, our segmentation model, with only **9.3M** parameters, attains an mIoU of **42.6%**, outperforming larger ANN and SNN baselines. These results further confirm the scalability of our design to dense prediction tasks.

## 6 CONCLUSION

We present *SpikeDet*, a fully spiking detector designed to achieve both precise regression and effective spatial semantic modeling in object detection with SNNs. As the foundation of our framework, the *TE-LIF* neuron bridges biologically inspired time-evolving membrane dynamics with the high-precision requirements of regression tasks, offering enhanced expressivity while remaining hardware-friendly. Complementing this design, the *Dual-Stream Spiking Attention* incorporates a QV-only architecture with parallel GlobalMixer and LocalAmplifier modules to effectively capture global context and local detail with linear complexity. Together, these components enable SpikeDet to set new state-of-the-art results in SNN-based detection, achieving **68.3% mAP@50** and **51.9% mAP@50:95** on COCO with an energy cost of only **32.4 mJ**, while also generalizing well to classification and segmentation tasks. These findings highlight SpikeDet as a promising foundation for deploying SNNs in more complex and challenging real-world applications.

## 7 REPRODUCIBILITY STATEMENT

Full experimental details are in subsection 5.1, Appendix B, Appendix C, and Appendix E. Complete source code will be released in the final version.

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

APPENDIX

## A   PROOF OF THEOREM 4.1

*Proof.*

**1. Introducing Notations.**

From the definition of TE-LIF neuron:

$$H_t = H_{t-1} + (\boldsymbol{W}\boldsymbol{x} + \boldsymbol{b})\omega_t - V_{th}\omega_t S_t$$

Rearranging yields

$$\frac{H_t}{\omega_t} - \frac{\omega_{t-1}}{\omega_t}\frac{H_{t-1}}{\omega_{t-1}} = (\boldsymbol{W}\boldsymbol{x} + \boldsymbol{b}) - V_{th}S_t.$$

Defining $\hat{H}_t = H_t/\omega_t$, we obtain

$$\hat{H}_t - \frac{\omega_{t-1}}{\omega_t}\hat{H}_{t-1} = (\boldsymbol{W}\boldsymbol{x} + \boldsymbol{b}) - V_{th}S_t.$$

Let $\hat{\beta} = \omega_{t-1}/\omega_t$. The dynamics with the introduced simplified notations imply:

$$\hat{\beta}^i \hat{H}_{t-1-i} = \hat{\beta}^{i+1}\hat{H}_{t-2-i} + \hat{\beta}^i(\boldsymbol{W}\boldsymbol{x} + \boldsymbol{b}) - V_{th}\hat{\beta}^i S_{t-1-i}.$$

Taking the sum for $i = 0, \ldots, t-2$, we obtain

$$\hat{H}_{t-1} = \sum_{i=0}^{t-2} \hat{\beta}^i(\boldsymbol{W}\boldsymbol{x} + \boldsymbol{b}) - V_{th}\sum_{i=0}^{t-2} \hat{\beta}^i S_{t-1-i}.$$

Therefore, the spike activation $S_t$ is determined by

$$S_t = \Theta\Big(\hat{\beta}\hat{H}_{t-1} + \boldsymbol{W}\boldsymbol{x} + \boldsymbol{b} - V_{th}\boldsymbol{1}\Big)$$

$$= \Theta\left(\sum_{i=0}^{t-1} \hat{\beta}^i(\boldsymbol{W}\boldsymbol{x} + \boldsymbol{b}) - V_{th}\Big(\boldsymbol{1} + \sum_{i=1}^{t-1} \hat{\beta}^i S_{t-i}\Big)\right).$$

For an arbitrary neuron $k \in [n_1]$ in the first hidden layer, this becomes

$$S_{k,t} = \Theta\left(\sum_{i=0}^{t-1} \hat{\beta}^i(\langle\boldsymbol{w}_k, \boldsymbol{x}\rangle + b_k) + \hat{\beta}^t \hat{H}_{k,0} - V_{th}\Big(1 + \sum_{i=1}^{t-1} \hat{\beta}^i S_{k,t-i}\Big)\right)$$

$$= \Theta\left(\langle\boldsymbol{w}_k, \boldsymbol{x}\rangle + b_k + \frac{\hat{\beta}^t \hat{H}_{k,0} - V_{th}\Big(1 + \sum_{i=1}^{t-1} \hat{\beta}^i S_{k,t-i}\Big)}{\sum_{i=0}^{t-1} \hat{\beta}^i}\right), \tag{8}$$

where $\boldsymbol{w}_i \in \mathbb{R}^{n_{\text{in}}}$ denotes the $i$-th row vector of $\boldsymbol{W}$. This means that at time step $t \in [T]$, the value of $S_{k,t} \in \{0, 1\}$ gives information about the half-space the input vector $\boldsymbol{x} \in \mathbb{R}^{n_{\text{in}}}$ lies in with respect to the hyperplane

$$h_{t-1}(S_{k,1}, \ldots, S_{k,t-1}) := \big\{\boldsymbol{x} \in \mathbb{R}^{n_{\text{in}}} : \langle\boldsymbol{w}_k, \boldsymbol{x}\rangle + b_k - g_{t-1}(S_{k,1}, \ldots, S_{k,t-1}) = 0\big\} \subset \mathbb{R}^{n_{\text{in}}}$$

where the function $g_{t-1}$ is defined by

$$g_{t-1} : \{0, 1\}^{t-1} \mapsto \mathbb{R}, \quad g_{t-1}(a_1, \ldots, a_{t-1}) = \frac{-\hat{\beta}^t \hat{H}_{k,0} + V_{th}\Big(1 + \sum_{i=1}^{t-1} \hat{\beta}^i a_{t-i}\Big)}{\sum_{i=0}^{t-1} \hat{\beta}^i}. \tag{9}$$

Furthermore, for each binary code $(a_i)_{i \in [t-1]} \in \{0, 1\}^{t-1}$, we define the corresponding region

$$R_{t-1}(a_1, \ldots, a_{t-1}) := \{\boldsymbol{x} \in \mathbb{R}^{n_{\text{in}}} : S_{k,i} = a_i \ \forall i \in [t-1]\} = \cap_{i=1}^{t-1} \{\boldsymbol{x} \in \mathbb{R}^{n_{\text{in}}} : S_{k,i} = a_i\}.$$

Note that such a region can be empty (see below) and we denote by $N(t)$ the number of non-empty such regions (which is also the total number of regions created at time step $t$). Our starting point is

the step $t = 1$, i.e., $t - 1 = 0$, where the whole space $\mathbb{R}^{n_{\text{in}}}$, which corresponds to the empty code $(a_i)_{i=1}^0$, is divided by exactly $2^0 = 1$ hyperplane, namely (according to (9)) the one given by the shift $g_0 = -\hat{\beta}\hat{H}_{k,0} + V_{th}$, into 2 different regions (depending on whether $a_1 = 0$ or $a_1 = 1$).

**2. Not Every Binary Code Corresponds to a Non-Empty Region.**

In principle, after time step $t-1$, or equivalently, before time step $t$, there can be $2^{t-1}$ possible binary codes $(a_i)_{i \in [t-1]} \in \{0, 1\}^{t-1}$ and accordingly the same number of hyperplanes $h_{t-1}(a_1, \ldots, a_{t-1})$. Each of these hyperplanes may separate (at most) one region into two sub-regions, thus increasing the total number of regions by one. This means that the number of regions might be doubled in each time step, i.e., $N(t) - N(t - 1)$ might reach $2^{t-1}$, which possibly leads to $1 + \sum_{t=1}^{T} 2^{t-1} = 2^T$ regions in total at time step $T$.

However, in reality, a hyperplane can divide a region into two sub-regions only if it intersects (in our case, as the hyperplanes are parallel, if it lies inside) that region, because otherwise the region remains one whole region. More specifically in our case, a region $R_{t-1}(a_1, \ldots, a_{t-1})$ corresponding to the code $(a_1, \ldots, a_{t-1})$ defined before time $t$ is separated into two sub-regions at time $t$ if and only if it contains the hyperplane $h_{t-1}(a_1, \ldots, a_{t-1})$ (created at time step $t$), i.e.,

$$h_{t-1}(a_1, \ldots, a_{t-1}) \subset R_{t-1}(a_1, \ldots, a_{t-1}). \tag{10}$$

According to our previous notion of (non-)empty regions, this means that if $h_{t-1}(a_1, \ldots, a_{t-1})$ falls outside of $R_{t-1}(a_1, \ldots, a_{t-1})$, i.e., the condition (10) is violated, then the whole region $R_{t-1}(a_1, \ldots, a_{t-1})$ must lie on one side of the hyperplane $h_{t-1}(a_1, \ldots, a_{t-1})$ and therefore either $R_t(a_1, \ldots, a_{t-1}, 0)$ or $R_t(a_1, \ldots, a_{t-1}, 1)$ is empty, while the other set is the same as $R_{t-1}(a_1, \ldots, a_{t-1})$.

The requirement (10) significantly reduces the number of separated regions, or equivalently, reduces the increase $N(t) - N(t - 1)$ in the number of regions from time step $t - 1$ to $t$.

**3. Deriving the Bound on $N(T)$.**

(1) For $\hat{\beta} \leq 1$:

We fix a time step $t \in [T]$ and consider the transition from $t - 1$ to $t$. Moreover, let $m \in \{0, \ldots, t - 1\}$ be arbitrary and consider the set

$$A_m := \{(a_i)_{i \in [t-1]} \in \{0, 1\}^{t-1} : \sum_{i=1}^{t-1} a_{t-i} = m \text{ and } R_{t-1}(a_1, \ldots, a_{t-1}) \neq \emptyset\}$$

of all binary codes of length $t - 1$ that have $m$ ones in their representation and correspond to a non-empty region created before time $t$. Observe that if we arrange the codes $(a_1, \ldots, a_{t-1})$ in $A_m$ in increasing lexicographic order, then the corresponding values $\sum_{i=1}^{t-1} a_{t-i}\hat{\beta}^i$ are in decreasing order (since $\hat{\beta}^i$ decreases with increasing $i$). This means that while the regions $R_{t-1}(a_1, \ldots, a_{t-1})$ are arranged in increasing lexicographic order of $(a_1, \ldots, a_{t-1})$ [1], the position of their corresponding hyperplanes $h_{t-1}(a_1, \ldots, a_{t-1})$ are arranged in the reversed order, i.e., in lexicographic order of $(a_{t-1}, \ldots, a_1)$. Since the regions are all disjoint, it follows that there is at most one hyperplane belonging to the 'correct' region, i.e., the region that corresponds to the same binary code. Since $m \in \{0, \ldots, t - 1\}$ was arbitrary, we deduce that there are at most $t$ hyperplanes belonging to the 'correct' regions at time step $t$. Hence, at the transition from time step $t - 1$ to $t$, we obtain

$$N(t) \leq N(t - 1) + t.$$

Taking the sum over $t \in [T]$, we get

$$N(T) \leq 1 + \sum_{t=1}^{T} t = 1 + \frac{T(T + 1)}{2} \in O(T^2).$$

---

[1] Intuitively, the new sub-region at any time step $i$ lies on the left of the hyperplane $h_{i-1}(a_1, \ldots, a_{i-1})$ if $a_i = 0$ and on the right if $a_i = 1$, and this process is performed from $i = 1$ on. The process actually reflects the ordering of binary codes in lexicographic order.

(2) For $\hat{\beta} > 1$:

When $\hat{\beta} > 1$, since the order of regions and the order of hyperplanes are no longer opposite, it seems that the upper bound could return to $O(2^T)$. However, because the distances between hyperplanes increase exponentially while the lengths of the feasible regions shrink over time, the number of new cuts is still limited. To formalize this, we refine the sets $A_m$ by defining

$$A_{m,n} := \Big\{ (a_i)_{i \in [t-1]} \in A_m : a_n = 1 \text{ and } a_j = 0, \ j = n+1, \ldots, T \Big\},$$

that is, $A_{m,n}$ collects all histories in which the last spike occurs exactly at time $n$. Fix a neuron in the first layer with weight vector $w$, and let $z = \langle w, x \rangle$ be the scalar projection of the input. Given a history code $a_{1:k} \in \{0,1\}^k$, the candidate threshold at step $k+1$ is

$$T_k(a_{1:k}) := \frac{V_{th}\Big(1 + \sum_{i=1}^k \hat{\beta}^{k+1-i} a_i\Big)}{\sum_{i=0}^k \hat{\beta}^i} - b. \tag{11}$$

At time $t-1$, the feasible interval is

$$I_{t-1}(a_{1:t-1}) = \Big[ \underbrace{\max_{k \le t-1 : a_k = 1} T_{k-1}(a_{1:k-1})}_{=:L(a)}, \ \underbrace{\min_{k \le t-1 : a_k = 0} T_{k-1}(a_{1:k-1})}_{=:U(a)} \Big). \tag{12}$$

For any $a, a' \in A_{m,n}$, the distance between their candidate hyperplane intercepts at step $t$ satisfies

$$\big|T_k(a) - T_k(a')\big| = \frac{V_{\text{th}}}{\sum_{i=0}^{t-1} \hat{\beta}^i} \Big| \sum_{r=0}^{n-2} \hat{\beta}^r \varepsilon_r \Big|, \qquad \varepsilon_r \in \{-1, 0, 1\}.$$

Definite $\Delta_{\min}^{(n)}(t) = \min_{a,a'} \big| \hat{H}_t(a) - \hat{H}_t(a') \big|$, when $\hat{\beta} = 2$,

$$\Delta_{\min}^{(n)}(t) \ \ge \ \frac{2^{t-n+1}}{2^t - 1} V_{\text{th}} \ \ge \ 2^{-(n-1)} V_{\text{th}}.$$

If $a \in A_{m,n}$ then $a_j = 1, a_{j+1} = \cdots = a_{t-1} = 0$. Since

$$L(a) = \max_{k : a_k = 1} T_{k-1}(a_{1:k-1}) \ge T_{j-1}(a_{1:j-1}), \quad U(a) = \min_{k : a_k = 0} T_{k-1}(a_{1:k-1}) \le T_j(a_{1:j}),$$

we always have

$$\text{width}\big(I_{t-1}(a)\big) = U(a) - L(a) \ \le \ T_j(a_{1:j}) - T_{j-1}(a_{1:j-1}). \tag{3}$$

Now define

$$S_j := \sum_{i=1}^{j-1} \hat{\beta}^{j-i} a_i, \quad D_{j-1} := \sum_{i=0}^{j-1} \hat{\beta}^i = \frac{\hat{\beta}^j - 1}{\hat{\beta} - 1}, \quad D_j := \sum_{i=0}^{j} \hat{\beta}^i = \frac{\hat{\beta}^{j+1} - 1}{\hat{\beta} - 1}.$$

A direct algebraic simplification yields

$$T_j(a_{1:j}) - T_{j-1}(a_{1:j-1}) = \frac{-V_{th} S_j \ + \ V_{th} \dfrac{\hat{\beta}(\hat{\beta}^{j-1} - 1)}{\hat{\beta} - 1}}{D_{j-1} D_j}. \tag{4}$$

Consequently, by setting $S_j \ge 0$ we obtain the explicit bound

$$\text{width}\big(I_{t-1}(a)\big) \ \le \ \frac{\hat{\beta}^j |\hat{H}_0| \ + \ V_{th} \dfrac{\hat{\beta}(\hat{\beta}^{j-1} - 1)}{\hat{\beta} - 1}}{D_{j-1} D_j}. \tag{5}$$

After simplification,

$$\text{width}\big(I_{t-1}(a)\big) \ \le \ \frac{(\hat{\beta} - 1)V_{th}}{\hat{\beta}^{j+1} - 1} \ + \ \frac{(\hat{\beta} - 1)^2 \, \hat{\beta}^j |\hat{H}_0|}{(\hat{\beta}^j - 1)(\hat{\beta}^{j+1} - 1)}. \tag{6}$$

In particular, for $\hat{\beta} = \omega_{t-1}/\omega_t = 2^{T-t+1}/2^{T-t} = 2$ in our experimental setting,

$$\text{width}\big(I_{t-1}(a)\big) \ \leq \ \frac{V_{th}}{2^{j+1}-1} \ + \ \frac{|\hat{H}_0|}{(2^j-1)(2^{j+1}-1)} \cdot 2^j. \tag{7}$$

Combining the spacing lower bound and the width upper bound, a packing argument gives

$$M_{i,j}(t) \ \leq \ \left\lceil \frac{\text{width}(I_{t-1}(a))}{\Delta_{\min}^{(j)}(t)} \right\rceil + 1.$$

For $\hat{\beta} = 2$, using (2) and (7) one finds

$$M_{i,j}(t) \ \leq \ 1 + \left\lceil \tfrac{1}{3} + \tfrac{|\hat{H}_0|}{4V_{th}} \right\rceil. \tag{8}$$

For general $\hat{\beta} > 1$, using (1) and (6),

$$M_{i,j}(t) \ \leq \ 1 + \left\lceil \frac{\hat{\beta}-1}{\hat{\beta}+1} + \frac{(\hat{\beta}-1)\,|\hat{H}_0|}{V_{th}(\hat{\beta}+1)} \right\rceil. \tag{9}$$

Summing over $j = 0, \ldots, t-1$ yields

$$\Delta N(t) \ = \ \sum_{j=0}^{t-1}\sum_{i=0}^{t-1} M_{i,j}(t) \ \leq \ C(\hat{\beta}, V_{th}, \hat{H}_0) \cdot t^2,$$

and consequently

$$N(T) \ \leq \ N(0) + \sum_{t=1}^{T} \Delta N(t) \ = \ O(T^3),$$

with an explicit constant $C$. For example, at $\hat{\beta} = 2$,

$$C(2, V_{th}, \hat{H}_0) = 1 + \left\lceil \tfrac{1}{3} + \tfrac{|\hat{H}_0|}{4V_{th}} \right\rceil.$$

$\square$

## B    VERILOG IMPLEMENTATION OF TE-LIF NEURON

Given that the complete Verilog code is excessively lengthy, we present a pseudocode version of the TE-LIF neuron's Verilog implementation for brevity and readability.

**TE-LIF Neuron, Verilog Pseudocode.**

```
Neuron TE-LIF {
    // Constants
    T = 8 // Time steps
    FRAC_BITS = 8 // Fixed-point fractional bits
    WEIGHTS = [32768, 16384, 8192, 4096, 2048, 1024, 512, 256]
    // Fixed-point weights: [128, 64, 32, 16, 8, 4, 2, 1]
    INIT_V = 128 // Initial voltage: 0.5 in fixed-point

    // State machine states
    IDLE = 0, LOAD_INPUT = 1, ACCUMULATE = 2,
    GENERATE_SPIKES = 3, UPDATE_VOLTAGE = 4, OUTPUT = 5

    // Internal registers
    state = IDLE
    v = 0 // Membrane potential (16-bit signed)
    x_seq[8] = {0} // Input sequence buffer
    y_seq[8] = {0} // Output spike sequence
    t_counter = 0 // Time step counter (0-7)
    current_weight = 0 // Current weight to subtract

    // Main state machine (triggered by clock rising edge)
    on clock_posedge:
        if reset:
            reset_all_registers()
        else:
            switch state:
                case IDLE:
                    if x_valid:
                        x_seq[0] = x_data // Load first input immediately
                        v = INIT_V // Initialize membrane potential
                        t_counter = 1
                        state = LOAD_INPUT

                case LOAD_INPUT:
                    if x_valid and t_counter < 8:
                        x_seq[t_counter] = x_data
                        t_counter = t_counter + 1
                    elif t_counter >= 8:
                        t_counter = 0
                        state = ACCUMULATE

                case ACCUMULATE:
                    if t_counter < 8:
                        // Weighted accumulation using bit shifts
                        switch t_counter:
                            case 0: v = v + (x_seq[0] << 7) // x * 128
                            case 1: v = v + (x_seq[1] << 6) // x * 64
                            case 2: v = v + (x_seq[2] << 5) // x * 32
                            case 3: v = v + (x_seq[3] << 4) // x * 16
                            case 4: v = v + (x_seq[4] << 3) // x * 8
                            case 5: v = v + (x_seq[5] << 2) // x * 4
                            case 6: v = v + (x_seq[6] << 1) // x * 2
                            case 7: v = v + x_seq[7] // x * 1
                        t_counter = t_counter + 1
                    else:
                        t_counter = 0
```

```
                    state = GENERATE_SPIKES

            case GENERATE_SPIKES:
                if t_counter < 8:
                    // Generate spike and record weight
                    if v >= WEIGHTS[t_counter]:
                        y_seq[t_counter] = 1
                        current_weight = WEIGHTS[t_counter]
                    else:
                        y_seq[t_counter] = 0
                        current_weight = 0
                    state = UPDATE_VOLTAGE
                else:
                    t_counter = 0
                    state = OUTPUT

            case UPDATE_VOLTAGE:
                v = v - current_weight // Update membrane potential
                t_counter = t_counter + 1
                if t_counter >= 8:
                    t_counter = 0
                    state = OUTPUT
                else:
                    state = GENERATE_SPIKES

            case OUTPUT:
                if t_counter < 8:
                    y_data = y_seq[t_counter] // Output current spike
                    y_valid = 1 // Assert output valid
                    t_counter = t_counter + 1
                else:
                    y_valid = 0 // Deassert output valid
                    state = IDLE // Return to idle state
    }
```

## C  FURTHER DISCUSSION ON DUAL-STREAM SPIKING ATTENTION

We present a comprehensive analysis of the proposed DSSA module, including its structural decomposition, computational complexity, and empirical efficiency.

Given the input spike map $X \in \{0,1\}^{N \times D}$, the spike-based Query and Value are computed as:

$$Q = \text{TE}(\text{BN}(XW_Q)), \quad V = \text{TE}(\text{BN}(XW_V)), \tag{13}$$

where $\text{TE}(\cdot)$ denotes the TE-LIF spiking neuron. Based on these representations, we construct a global-local attention mechanism formulated as:

$$Attn = \underbrace{\text{MLP}\left(\text{TE}\left(\sum_c Q\right)\right)}_{\text{GlobalMixer}} + \underbrace{\text{DepthWiseConv}(Q)}_{\text{LocalAmplifier}}, \quad X' = \text{TE}(Attn \odot V), \tag{14}$$

where $\odot$ denotes element-wise multiplication.

In the *GlobalMixer* module, a column-wise summation is first applied to $Q$, resulting in a compact binary representation $H \in \mathbb{R}^{1 \times D}$, with a computational cost of $O(ND)$. The resulting vector $H$ is subsequently passed through a two-layer bottleneck MLP:

$$Attn_{\text{GM}} = \text{Linear}_{D//r \to D}\left(\text{TE}\left(\text{BN}\left(\text{Linear}_{D \to D//r}(H)\right)\right)\right), \tag{15}$$

where $r$ is the hidden dimension reduction ratio. Here, $\text{Linear}_{A \to B}$ denotes a linear transformation from dimension $A$ to $B$. The overall complexity of this MLP is $O(D^2//r)$, which can be approximated as $O(ND)$ under the assumption that $D//r$ is much less than $N$ in practical scenarios. The output $Attn_{\text{GM}}$ is then broadcast to match the spatial dimension, resulting in a tensor of shape $\mathbb{R}^{N \times D}$.

The *LocalAmplifier* component performs depthwise convolution across channels, extracting local spatial features in a channel-wise manner. This operation also incurs a computational complexity of $O(ND)$, and produces an output $Attn_{\text{LA}} \in \mathbb{R}^{N \times D}$.

Finally, the broadcasted global attention $Attn_{\text{GM}}$ is combined with the local attention $Attn_{\text{LA}}$ via element-wise addition to yield the final attention map $Attn$, leading to a complexity of $O(ND)$. This map is then applied to modulate the binary tensor $V$ through element-wise multiplication, which can be regarded as mask operations without energy cost.

In summary, the proposed attention mechanism achieves a total computational complexity of $O(ND)$, attributed to its lightweight architectural design and the elimination of costly matrix multiplications. Additionally, by removing the Key ($K$) branch, the model reduces both computational and parameter overhead, thereby improving efficiency in terms of both computation and memory usage. From a practical standpoint, on the COCO dataset, a single inference of DSSA consumes only **36%** of the power required by SDSA-3 (Yao et al., 2024), a prevalent attention mechanism in the SNN field with a higher complexity of $O(ND^2)$.

# D DATASETS DETAILS

## D.1 OBJECT DETECTION

We evaluated the proposed SpikeDet model on four object detection datasets, covering conventional, neuromorphic, and remote sensing domains:

**COCO** (Lin et al., 2014): The *Common Objects in Context* (COCO) dataset is a predominant benchmark for object detection, comprising 80 object categories with 118,000 training images and 5,000 validation images. It provides complex scenes with multiple objects, occlusions, and varied scales, serving as a standard for model comparison.

**Gen1** (De Tournemire et al., 2020): The *Gen1* dataset is a large-scale neuromorphic benchmark tailored for object detection. It includes 39 hours of real-world driving data captured by an ATIS event-based camera, offering asynchronous event streams. The dataset provides over 255,000 manually annotated bounding boxes for pedestrians and vehicles, enabling evaluation in event-driven vision settings.

**NWPU VHR-10** (Cheng et al., 2017): This dataset contains very high-resolution (VHR) optical remote sensing images across 10 object categories: airplanes, ships, storage tanks, baseball diamonds, tennis courts, basketball courts, ground track fields, harbors, bridges, and vehicles. The challenging imagery includes variations in scale, orientation, and background complexity. In our experiments, we randomly split the dataset into training and validation sets following a 70%:30% ratio.

**SSDD** (Wang et al., 2019): The *SAR Ship Detection Dataset* (SSDD) focuses on ship detection using Synthetic Aperture Radar (SAR) images. It provides annotated ship instances under various sea states and imaging conditions, making it ideal for assessing detection performance in radar-based scenarios.

## D.2 IMAGE CLASSIFICATION

**ImageNet-1K** (Deng et al., 2009): The *ImageNet Large Scale Visual Recognition Challenge* (ILSVRC) 2012 dataset, commonly referred to as ImageNet-1K, contains over 1.2 million training images and 50,000 validation images spanning 1,000 object categories. It is one of the most widely used benchmarks for evaluating large-scale image classification models due to its diversity and scale.

## D.3 SEMANTIC SEGMENTATION

**ADE20K** (Zhou et al., 2017): The *ADE20K* dataset is a comprehensive benchmark for semantic segmentation, consisting of over 25,000 images covering a wide range of indoor and outdoor scenes. Each image is densely annotated with pixel-level labels across 150 semantic categories. Its complexity and diversity make it a standard dataset for evaluating scene understanding capabilities.

# E    EXPERIMENTAL DETAILS

## E.1    OBJECT DETECTION

Our object detection experiments are conducted on the macro architecture of YOLOv12 (Tian et al., 2025), chosen for its efficiency and built-in attention mechanisms, which provide a strong basis for accurate real-time detection.

To enable spike-driven computation, we make principled modifications aligned with our framework. Specifically, we replace the original activation functions with TE-LIF neurons (subsection 4.1), allowing the network to perform bio-plausible temporal dynamics. Based on this, we convert key components, such as convolutions and MLPs, into their spiking counterparts to fully support spike-driven computation. Additionally, since the vanilla attention in YOLOv12 depends on floating-point matrix operations and softmax, we substitute it with our lightweight Dual-Stream Spiking Attention module (subsection 4.2), which eliminates QK-based interactions, reduces complexity, and enables efficient global-local feature fusion.

The complete training configurations for object detection across the COCO, NWPU, and SSDD datasets are summarized in Table 5. All experiments are conducted on NVIDIA RTX 4090 GPUs. For the COCO dataset, we evaluate four model scales—S (Small), M (Medium), L (Large), and X (Extra Large)—to analyze performance under varying model capacities. For the NWPU and SSDD datasets, we adopt the M (Medium) model to evaluate performance. Data augmentation strategies include horizontal flipping with a probability of 0.5, while vertical flipping is disabled. Mosaic augmentation is applied with full probability, combining four images into one to enrich contextual understanding. Copy-Paste augmentation is employed in conjunction with horizontal flipping to enhance object diversity. Additionally, RandAugment is used as an automated data augmentation method. To increase robustness to occlusion, we apply random erasing with a probability of 0.4.

Table 5: Training configurations for object detection on the COCO, NWPU, and SSDD datasets.

| Setting | COCO(Lin et al., 2014) | | | | NWPU(Cheng et al., 2017) | SSDD(Wang et al., 2019) |
|---|---|---|---|---|---|---|
| | S | M | L | X | M | M |
| Param (M) | 9.5 | 21.8 | 31.8 | 71.2 | 21.8 | 21.8 |
| Batch size | 128 | 128 | 128 | 64 | 32 | 128 |
| Resolution | $640 \times 640$ | | | | $1024 \times 1024$ | $640 \times 640$ |
| Training epochs | 600 | | | | 300 | 300 |
| Learning rate | 0.01 | | | | 0.0007 | 0.02 |
| Optimizer | SGD | | | | AdamW | AdamW |
| Number of GPUs | 8 | | | | 4 | 4 |

For the Gen1 dataset, we train the medium-scale (21.8M) SpikeDet model on 2 NVIDIA A100 (80GB) GPUs, mainly adopting the configuration in SpikeYOLO (Luo et al., 2024). Each training sample consists of a 2.5-second event stream preceding the annotation, divided into 4 slices as input. The model is trained for 50 epochs with a batch size of 160 and a resolution of 320×320, using SGD with an initial learning rate of 0.02 decaying to 0.004. A 10-epoch warmup is applied, with a momentum of 0.8 and a bias learning rate of 0.1. All data augmentation strategies are disabled.

## E.2    IMAGE CLASSIFICATION

For image classification, we adopt the macro-architecture of E-SpikeFormer(Yao et al., 2025a).

We replace the original Spike Firing Approximation strategy in E-SpikeFormer with our proposed TE-LIF neuron, and substitute the Efficient Spike-Driven Self-Attention (E-SDSA) module with our newly designed lightweight Dual-Stream Spiking Attention. The resulting model contains only 7.8M parameters.

The model is trained on the ImageNet-1K (Deng et al., 2009) dataset using 8 NVIDIA RTX 4090 GPUs for 300 epochs with a batch size of 256. We adopt the AdamW optimizer with an initial learning rate of 3e-4 and a linear warm-up over the first 5 epochs. Data augmentation techniques include label smoothing, RandAugment, and random erasing. Weight decay and a cosine learning rate decay schedule are also applied during training.

### E.3 Semantic Segmentation

Semantic segmentation experiments are conducted on the ADE20K(Zhou et al., 2017) dataset using an encoder-decoder architecture. The encoder is based on E-SpikeFormer enhanced with our TE-LIF neuron and DSSA, while the decoder adopts a Query-based Feature Pyramid Network (QFPN). The model is initialized with ImageNet-pretrained weights and has 9.3M parameters.

Training is performed for 240k iterations on 8 NVIDIA RTX 4090 GPUs using automatic mixed precision (AMP). Optimization is carried out with the AdamW optimizer, an initial learning rate of 0.001, and a weight decay of 0.005. We apply a linear warm-up for the first 1,500 iterations followed by a polynomial learning rate decay. The input resolution is set to $512 \times 512$, and the batch size is 16. The data augmentation pipeline includes random resizing, random cropping, horizontal flipping, and photometric distortion. The segmentation loss is based on cross-entropy.

# F    ENERGY CONSUMPTION CALCULATION

To estimate energy consumption, we adopt a widely recognized evaluation protocol in the SNN community (Horowitz, 2014; Yao et al., 2023a; 2024; 2025b; Luo et al., 2024). This protocol ignores specific hardware implementation details and estimates the theoretical energy consumption of a given model, thus facilitating the quantitative energy evaluation across different SNN and ANN algorithms.

Under this protocol, the energy consumption of ANN is calculated as:

$$E_{ANN} = FL \cdot E_{\mathrm{MAC}} \tag{16}$$

where $FL$ denotes the total floating-point operations (FLOPs) required by the network and $E_{\mathrm{MAC}} = 4.6\mathrm{pJ}$ represents the energy cost of a single Multiply-and-Accumulate (MAC) operation in 45 nm technology (Horowitz, 2014).

In contrast, the energy consumption of the $n$-th layer in SNN is given by:

$$E_n = FL_n \cdot E_{\mathrm{AC}} \cdot fr_n \cdot T \tag{17}$$

where $FL_n$ is the number of FLOPs in the $n$-th layer, $fr_n$ is the average firing rate of that layer, $E_{\mathrm{AC}} = 0.9\,\mathrm{pJ}$ denotes the energy cost of an Accumulate (AC) operation in 45 nm technology (Horowitz, 2014), and $T$ is the timestep count. The overall energy consumption of the SNN is obtained by summing the energy consumption across all layers.

To support Equation 16 and Equation 17, the FLOPs for used layers are defined as follows. For a convolutional layer, the FLOPs are calculated as:

$$FL_{\mathrm{Conv}} = k^2 \cdot h_{\mathrm{out}} \cdot w_{\mathrm{out}} \cdot c_{\mathrm{in}} \cdot c_{\mathrm{out}} \, / \, g, \tag{18}$$

where $k$ is the kernel size, $(h_{\mathrm{out}}, w_{\mathrm{out}})$ denotes the height and width of the output feature map, $c_{\mathrm{in}}$ and $c_{\mathrm{out}}$ represent the number of input and output channels respectively, and $g$ represents the number of groups in grouped convolution.

Similarly, the FLOPs for a linear layer are computed as:

$$FL_{\mathrm{Linear}} = d_{\mathrm{in}} \cdot d_{\mathrm{out}}, \tag{19}$$

where $d_{\mathrm{in}}$ and $d_{\mathrm{out}}$ are the input and output dimensions of the linear layer, respectively.

# G  RESULTS ON EXTENDED TASKS

## G.1  IMAGENET CLASSIFICATION

Table 6: Performance of classification on ImageNet.

| Type | Model | Param(M) | Acc(%) |
|---|---|---|---|
| ANN | FastViT (Vasu et al., 2023) | 6.8 | 79.1 |
| | ResNet-50 (Li et al., 2021) | 25.6 | 79.4 |
| | GTP-DeiT (Xu et al., 2024) | 86.0 | 79.5 |
| ANN2SNN | FAST-SNN (Hu et al., 2023) | 138.4 | 73.0 |
| | Optimal (Bu et al., 2021) | 138.4 | 74.3 |
| | Two-stage (Wang et al., 2023b) | 138.4 | 74.9 |
| | TPP SNN (Bojković et al., 2025) | 22.0 | 77.8 |
| | MST (Wang et al., 2023c) | 28.5 | 78.5 |
| | ECMT (Huang et al., 2024) | 86 | 79.4 |
| Directly Trained SNN | QP-SNNs (Wei et al., 2025) | 13.3 | 61.4 |
| | TET-ResNet (Deng et al., 2022) | 21.8 | 68.0 |
| | SEW-ResNet (Fang et al., 2021) | 60.2 | 69.2 |
| | ReverB-SNN (Guo et al., 2025) | 21.8 | 70.9 |
| | Spikformer (Zhou et al., 2022) | 66.3 | 74.8 |
| | MS-ResNet (Hu et al., 2024) | 77.3 | 75.3 |
| | SNN-ViT (Wang et al., 2025) | 30.4 | 76.9 |
| | Att-MS-ResNet (Yao et al., 2023b) | 78.4 | 77.1 |
| | $\alpha$-SSA-Swin (Xiao et al., 2025) | 31.8 | 77.9 |
| | STAtten(SDT) (Lee et al., 2024) | 66.34 | 78.1 |
| | E-Spikeformer (Yao et al., 2025a) | 10.0 | 78.5 |
| | SpikingResformer (Shi et al., 2024) | 60.4 | 78.7 |
| | SpikformerV2 (Zhou et al., 2024b) | 29.11 | 78.8 |
| | QKFormer (Zhou et al., 2024a) | 16.47 | 78.8 |
| | **Ours** | 7.8 | **79.7** |

## G.2  ADE20K SEMANTIC SEGMENTATION

Table 7: Performance of semantic segmentation on ADE20K.

| Type | Model | Param(M) | mIoU(%) |
|---|---|---|---|
| ANN | ResNet-18 (Yu et al., 2022) | 15.5 | 32.9 |
| | PVT-Tiny (Wang et al., 2021) | 17.0 | 35.7 |
| | PVT-Small (Wang et al., 2021) | 28.2 | 39.8 |
| SNN | PSSD (Wang et al., 2024b) | - | 29.1 |
| | Meta-SpikeFormer (Yao et al., 2024) | 59.8 | 35.3 |
| | QSD-Transformer (Qiu et al., 2025) | 9.6 | 40.5 |
| | E-Spikeformer (Yao et al., 2025a) | 11.0 | 41.4 |
| | **Ours** | 9.3 | **42.6** |

## H LIMITATION

Due to limited resources, we leave the application of our method to language models and other domains for future work. While current deployments are limited to GPUs, we anticipate that its advantages will be more pronounced when implemented on neuromorphic hardware.

## I LARGE LANGUAGE MODEL USAGE STATEMENT

In accordance with the ICLR 2026 Author Guidelines on the use of large language models, we acknowledge that LLMs were utilized to refine phrasing and expression during the manuscript preparation. However, all scientific ideas, algorithmic designs, and experimental results are solely the work of the authors.

