# OpenReview forum: "Unleashing SNNs in Object Detection with Time-Evolving Neuron and Dual-Stream Spiking Attention"
_ICLR.cc/2026/Conference — ICLR 2026 Conference Withdrawn Submission_

### Official Review · Reviewer_KKZa · 2025-10-30

**Soundness:** 3
**Presentation:** 3
**Contribution:** 3
**Rating:** 6
**Confidence:** 5

**Summary:**

The paper proposes SpikeDet, a fully spiking object detector that combines precise regression with effective spatial semantic modeling for SNNs. The contributions include the TE-LIF neuron, which balances biologically inspired temporal dynamics with high-precision regression, and the Dual-Stream Spiking Attention, which captures both global context and local details efficiently. Experiments show state-of-the-art performance for SNN-based detection on multiple datasets with low energy consumption. Overall, SpikeDet presents a strong foundation for deploying SNNs in real-world tasks.

**Strengths:**

1. The topic of energy-efficient SNN for object detection is very interesting and import in neuromorphic computing community.

2. The proposed SpikeDet is thoroughly validated across multiple datasets and compared against SOTA methods, demonstrating clear overall advantages in accuracy and power consumption.

3. The writing is clear and easy to understand.

**Weaknesses:**

1. The authors assess the energy efficiency of SNNs primarily via simulated AC/MAC energy calculations rather than actual measurements on neuromorphic hardware. While this is common in the SNN field, directly reporting 32.4 mJ (Line 103) may be misleading. It would be more appropriate to report the relative energy efficiency compared to a corresponding ANN, e.g., how many times more efficient the SNN simulation is.

2. In Table 1, the simulated energy consumption of many ANN methods seems questionable. It is difficult to ensure the accuracy across so many architectures. I suggest focusing only on a few open-source SNN baselines for simulated energy comparison rather than all ANNs.

3. The manuscript presents two major modules (TE-LIF and Dual-Stream Spiking Attention), but no ablation study is shown to isolate each module’s contribution to overall detection performance. The authors should include this analysis.

4. In Table 3, the authors explore the impact of neuron types and time steps together. It is recommended to separate these analyses, with one table per conclusion, to improve clarity.

5. SNNs are generally suited for event-based data processing. While adding remote sensing datasets is interesting, it may not be particularly compelling for reviewers. In the future, focus could be on event-based object detection visualizations instead.

6. Minor suggestion: Combining event streams with RGB frames or other modalities is an emerging trend and may better reflect practical applications. The authors could reference relevant works [1-3] and briefly discuss how SpikeDet could be extended to handle multimodal data in future work.

[1] HDI-former: Hybrid dynamic interaction ANN-SNN transformer for object detection using frames and events, arXiv 2024.

[2] SODFormer: Streaming object detection with transformer using events and frames, TPAMI 2023.

[3] Retinomorphic object detection in asynchronous visual streams, AAAI 2022.

**Questions:**

Please response each comment in the weaknesses.

---

### Official Review · Reviewer_cE3c · 2025-10-31

**Soundness:** 3
**Presentation:** 3
**Contribution:** 2
**Rating:** 4
**Confidence:** 4

**Summary:**

This paper proposes a biologically-inspired neuron, TE-LIF, which achieves significant results on object detection tasks. Inspired by the dynamic propagation process in the hippocampus, this neuron attempts to enhance feature perception capabilities by applying different weights to activations at different time steps, thereby addressing the inherent sparse representation problem in SNNs. Additionally, the paper introduces a global and local weighted attention mechanism based on QV attention, which also effectively improves object detection performance. If my concerns are adequately addressed, I am willing to increase my rating.

**Strengths:**

(1) The biologically-inspired approach demonstrates a certain degree of innovation;

(2) The task extension experiments are comprehensive, and the supplementary materials provide detailed testing that validates the effectiveness of the proposed method;

(3) The theoretical analysis is thorough and reproducibility is well-supported.

**Weaknesses:**

(1) The connection between the proposed TE-LIF and object detection tasks is relatively weak;

(2) Comparisons with state-of-the-art SNN-based object detection methods are lacking;

(3) In the extension tasks, it is unclear whether the performance improvements stem from TE-LIF or the attention mechanism;

(4) The proposed attention mechanism appears to be an incremental integration of existing methods;

(5) A pipeline diagram is absent;

(6) The ablation studies are insufficiently comprehensive and the analysis is relatively weak.

**Questions:**

(1) TE-LIF appears to process information through weighting along the temporal dimension. The connection to the hippocampal dynamic observation described in the paper seems tenuous. Please provide further clarification;

(2) What distinguishes this attention mechanism from existing known methods? What is its relevance to the core task of object detection?

(3) Could the authors provide a pipeline diagram?

(4) The authors claim that TE-LIF enhances the perception capability and efficiency of SNNs. Should this be substantiated based on inference frame rates?

(5) Could the authors provide additional training details for the comparison methods to demonstrate the fairness of the comparative experiments?

---

### Official Review · Reviewer_zota · 2025-10-31

**Soundness:** 2
**Presentation:** 2
**Contribution:** 2
**Rating:** 2
**Confidence:** 5

**Summary:**

To narrow the gap with ANN-based object detection methods, this paper proposes SpikeDet, a fully spiking object detector that redefines both the microscopic neuron model and macroscopic attention mechanism. At its core, the bio-inspired TE-LIF neuron, with time-evolving membrane dynamics, enhances representational precision and achieves finer input pattern recognition, while maintaining computational efficiency.

**Strengths:**

1. This paper presents a very substantial amount of work.
2. The results are excellent; SpikeDet attains 68.3% mAP@50 and 51.9% mAP@50:95 on COCO.

**Weaknesses:**

1. I have significant difficulty understanding how the proposed TE-LIF neuron enhances the model's expressive power. As I interpret it, this mechanism primarily amplifies information at the early timesteps while progressively reducing the influence of later timesteps. This characteristic seems extremely detrimental for event-based camera modalities, which rely on temporal information. Moreover, even for standard RGB inputs, the utility of this approach is unclear.
If a neuron fires continuously across all timesteps, a standard LIF neuron already struggles to distinguish whether this is due to a large input at the first timestep or sustained large inputs at every timestep. The proposed TE-LIF neuron would only exacerbate this ambiguity.
However, it is puzzling that the experimental results are reportedly very strong. I find this highly counter-intuitive. I request the authors to provide a thorough and reasonable explanation for this discrepancy and to release the corresponding code for verification. I believe these experimental findings are highly unusual.
I have carefully examined the hyperparameters published by the authors and noted the model was trained for 600 epochs. Could this extensive training be the reason for the performance? Even so, it seems unlikely to account for such a significant improvement.
Furthermore, the I-LIF neuron [1] is currently considered one of the most powerful SNN neurons. What specific problems does TE-LIF solve that I-LIF cannot? Is there any experimental comparison between the two? If the motivation is to handle varying information importance across timesteps, an input encoding approach, such as [2], would seem more appropriate.

2. I am also unclear on the specific function of the proposed Dual-Stream Spiking Attention. The authors have not clearly specified its position within the model architecture. What problem does its inclusion solve? Why was it chosen for combination with TE-LIF over other attention mechanisms? Furthermore, is the comparison against other attention mechanisms in the ablation study a fair one?

3. The narrative of the paper feels disorganized. The introduction begins by analyzing two parallel problems SNNs face in object detection. However, the fourth paragraph of the introduction pivots, positioning TE-LIF as the core contribution. This is confusing. What is the central thesis of this paper?

4. Regarding experimental fairness: The authors compare their method against SpikeYOLO, which they identify as a strong baseline. However, SpikeYOLO was run with T=4 (timesteps), while the authors' model uses T=8. This is not a fair comparison.

5. This paper presents a method for object detection, yet it includes experiments on classification and segmentation. It is common to see a classification method evaluated for its generalization capabilities on downstream tasks like detection and segmentation, but the reverse (a detection-focused method being tested on classification) is unusual, and its motivation is unclear.

[1] Yao M, Qiu X, Hu T, et al. Scaling spike-driven transformer with efficient spike firing approximation training[J]. IEEE Transactions on Pattern Analysis and Machine Intelligence, 2025.
[2] Qiu X, Zhu R J, Chou Y, et al. Gated attention coding for training high-performance and efficient spiking neural networks[C]//Proceedings of the AAAI conference on artificial intelligence. 2024, 38(1): 601-610.

**Questions:**

1. The specific model configurations and code for the method in this paper have not been made public. I believe this presents a problem for reproducibility.
2. See Weaknesses part.

---

### Official Review · Reviewer_Un6D · 2025-11-10

**Soundness:** 2
**Presentation:** 2
**Contribution:** 2
**Rating:** 4
**Confidence:** 4

**Summary:**

This paper proposes SpikeDet, a fully spiking object detector that introduces two main contributions: (1) TE-LIF neurons with time-evolving membrane dynamics using power-of-two temporal weights, and (2) Dual-Stream Spiking Attention (DSSA) that removes matrix multiplication and employs QV-only design. The method achieves 68.3% mAP@50 and 51.9% mAP@50:95 on COCO with 32.4 mJ energy consumption.

**Strengths:**

1. Well-motivated problem: The paper clearly identifies two key limitations of SNNs in object detection - precision bottleneck and limited spatial modeling - and addresses both systematically.
2. Theoretical justification: Theorem 4.1 provides theoretical analysis showing TE-LIF achieves O(T³) region partitioning vs O(T²) for LIF, though the proof could be more accessible.
3. Comprehensive ablations: The ablation studies are thorough, examining the impact of neuron types, timesteps, and attention components.

**Weaknesses:**

1. The time-decaying weight ωt = 2^(T-t) appears somewhat arbitrary. Why specifically powers of two in decreasing order? Have you explored other temporal weighting schemes (e.g., learned weights, exponential decay, linear decay)?
2. Practical relevance: The O(T³) vs O(T²) complexity difference may not be significant for the small T values used (T=3-8). Table 3 shows performance saturates at T=8, limiting practical impact.
3. While you cite Wang et al., 2025 and Xiao et al., 2025 about QK struggles under sparse spikes, this deserves deeper analysis. What specifically fails about QK similarity under spike coding?
4. You use "multi-bit training" (merging timesteps, integer-valued spikes) but binary spikes at inference. This training-inference mismatch could be problematic.
5. No analysis of sensitivity to key hyperparameters (e.g., decay factor β in LIF neurons, MLP hidden dimension ratio r in DSSA)

**Questions:**

1. In Eq. 4, you show WI_t^l = (W·S_t^{l-1})·ωt. But during training with integer values, how exactly do you handle the ωt scaling? Is it applied to gradients as well?
2. You claim O(ND) complexity, but the MLP in GlobalMixer is O(D²/r). For D>>N or small r, this could dominate. What are typical D, N, r values in your experiments?
3. You test on frame-based (COCO, NWPU, SSDD) and event-based (Gen1) data. But Gen1 is converted to frames. Have you tested on true event streams?

---

### Note · Authors · 2025-11-14

I have read and agree with the venue's withdrawal policy on behalf of myself and my co-authors.